# The Biological Bases of Group 2 Pulmonary Hypertension

**DOI:** 10.3390/ijms20235884

**Published:** 2019-11-23

**Authors:** Ana I. Fernández, Raquel Yotti, Ana González-Mansilla, Teresa Mombiela, Enrique Gutiérrez-Ibanes, Candelas Pérez del Villar, Paula Navas-Tejedor, Christian Chazo, Pablo Martínez-Legazpi, Francisco Fernández-Avilés, Javier Bermejo

**Affiliations:** 1Department of Cardiology, Hospital General Universitario Gregorio Marañón, 28007 Madrid, Spain; anaferavi@gmail.com (A.I.F.); ryotti@gmail.com (R.Y.); anagmansilla80@gmail.com (A.G.-M.); teresamombiela@gmail.com (T.M.); egutibanes@gmail.com (E.G.-I.); cperezdelvillar@gmail.com (C.P.d.V.); paulanavastejedor@gmail.com (P.N.-T.); christianchazopaz@gmail.com (C.C.); legazpi.pablo@gmail.com (P.M.-L.); faviles@secardiologia.es (F.F.-A.); 2Instituto de Investigación Sanitaria Gregorio Marañón, 28007 Madrid, Spain; 3Centro de Investigación Biomédica en Red, CIBERCV, Instituto de Salud Carlos III, 28026 Madrid, Spain; 4Facultad de Medicine, Universidad Complutense de Madrid, 28007 Madrid, Spain

**Keywords:** pulmonary hypertension group 2, left heart disease, isolated pulmonary hypertension, combined pulmonary hypertension, gene, epigenetics

## Abstract

Pulmonary hypertension (PH) is a potentially fatal condition with a prevalence of around 1% in the world population and most commonly caused by left heart disease (PH-LHD). Usually, in PH-LHD, the increase of pulmonary pressure is only conditioned by the retrograde transmission of the left atrial pressure. However, in some cases, the long-term retrograde pressure overload may trigger complex and irreversible biomechanical and biological changes in the pulmonary vasculature. This latter clinical entity, designated as combined pre- and post-capillary PH, is associated with very poor outcomes. The underlying mechanisms of this progression are poorly understood, and most of the current knowledge comes from the field of Group 1-PAH. Treatment is also an unsolved issue in patients with PH-LHD. Targeting the molecular pathways that regulate pulmonary hemodynamics and vascular remodeling has provided excellent results in other forms of PH but has a neutral or detrimental result in patients with PH-LHD. Therefore, a deep and comprehensive biological characterization of PH-LHD is essential to improve the diagnostic and prognostic evaluation of patients and, eventually, identify new therapeutic targets. Ongoing research is aimed at identify candidate genes, variants, non-coding RNAs, and other biomarkers with potential diagnostic and therapeutic implications. In this review, we discuss the state-of-the-art cellular, molecular, genetic, and epigenetic mechanisms potentially involved in PH-LHD. Signaling and effective pathways are particularly emphasized, as well as the current knowledge on -omic biomarkers. Our final aim is to provide readers with the biological foundations on which to ground both clinical and pre-clinical research in the field of PH-LHD.

## 1. Introduction

Pulmonary hypertension (PH) is a major health issue. Although epidemiological studies are still scarce, it is estimated that the annual prevalence of PH is around 1% of the world population and is probably increasing [1]. In Western countries, it could reach up to 6% for subjects over 85 years old [1], and it is more frequent in women than men [2]. Understanding the biological foundations of the disease is therefore of major interest.

Pulmonary hypertension is defined by a mean pulmonary artery pressure (mPAP) at rest of 25 mm Hg or more, according to established guidelines [3], or 20 mm Hg or more, according to reviewed criteria [4]. The multiple conditions that may cause PH are clustered into five groups [5]. Group 1-PH—also designated pulmonary arterial hypertension (PAH)—accounts for heritable, idiopathic, drug or HIV-related causes among others. Group 3-PH accounts for PH associated with chronic lung diseases, Group 4-PH for pulmonary artery remodeling due to major vessel thromboembolism, whereas Group 5-PH designates a heterogeneous group of multifactorial and unclear mechanisms.

Group 2-PH—PH due to left heart disease (PH-LHD)—designates PH caused by the retrograde transmission of left-atrial (LA) pressure to the pulmonary circulation. This is the source for roughly half of the cases of PH [1]. The diagnosis of PH-LHD is established when LA pressure (or its surrogate, the pulmonary capillary wedge pressure) is above 15 mmHg. Atrial pressure overload is caused by either systolic or diastolic left-ventricular dysfunction and/or valvular heart disease (VHD). In most cases of PH-LHD, the increase in pulmonary pressure is purely “passive” and is conditioned by the retrograde transmission of LA pressure—isolated post-capillary PH (iPC-PH). However, in some cases of PH-LHD, complex biomechanical and biological changes take place in the pulmonary vasculature, which may persist even when the left-heart predisposing condition is solved. The hallmark sign of this coexistent pre-capillary component is an increase in the transpulmonary pressure gradient (TPPG, the pressure difference between the mean pulmonary arterial pressure and PCWP) and in the pulmonary vascular resistance (PVR, the TPPG divided by cardiac output). When PVR is ≥3 Wood units, the clinical entity is designated as combined pre and postcapillary PH (cPC-PH) [6]. Differentiating iPC- from cPC-PH is of major clinical interest because they have a different prognosis and response to treatment [7,8]. Furthermore, the finding that patients with cPC-PH may share phenotypic fingerprints with PAH [9] has recently propelled interest in the biological foundations of PH-LHD.

Valvular heart disease (VHD) is a particularly well-suited condition for research in the field of PH-LHD. As opposed to the other causes of PH-LHD, in VHD, the initiating cardiac lesion is amenable to complete hemodynamic restoration. However, the natural history studies of this disease have demonstrated that PH may persist or even develop when the source of retrograde pressure transmission has been corrected [10]. This suggests that impaired molecular and biomechanical pathways sometimes are not re-established once dysregulated for a certain amount of time. However, there is a huge variability in the response of the lung vasculature to the hemodynamic overload and its correction. Unfortunately, the source of such variability is still mostly ununderstood.

All multicentric clinical trials targeting the lung vasculature in PH-LHD have been negative. Investigators recognize that mechanisms of plastic remodeling of the pulmonary circulation in this entity need to be clarified to adequately stratify patient outcomes and successfully identify new therapeutic targets. In the following sections, we review the current state-of-the art biological aspects known to be involved in PH-LHD, emphasizing their potential clinical and therapeutic implications.

## 2. The Hemodynamic and Structural Changes of PH-LHD

The rise in backward pressure at the left atrium and the pulmonary veins is the hemodynamic hallmark of iPC-PH. This upstream pressure transmission has both steady and pulsatile-flow consequences on the pressure of the pulmonary artery, which are only beginning to be understood [11,12]. Vascular tone at this level is regulated by the intimal (endothelial), and medial (vascular smooth muscle) layers and the interlayer interactions, which respond to differentiated stimuli, such as pressure-dependent responses, neural mechanisms, hormonal mechanisms, local metabolic regulations, and endothelial products [13]. The classical paradigm for PH-LHD assumes that longstanding venous hypertension induces structural alterations of the pulmonary arterial bed that characterize cPC-PH [14]. Medial hypertrophy and intimal proliferation of the distal pulmonary arteries and arterioles dramatically raise PVR [15]. Interstitial lung edema and alveolar hemorrhage is frequently found, and the vasomotor tone of the pulmonary arteries is also increased. Hypertrophic remodeling of distal pulmonary arteries, fibrosis, and luminal occlusion are likewise common findings in PH-LHD and PAH, but the typical plexiform lesions of PAH are absent [16,17]. As in any type of PH, the increased vascular load leads to right ventricular diastolic dysfunction [18] and afterload mismatch, eventually causing right ventricle dilation, failure, congestive symptoms, and reduced survival [19].

Recent morphological studies are changing this paradigm. A comprehensive analysis of necropsy lung specimens of patients with PH-LHD has identified not only remodeled arterial vessels but also marked thickening of both pulmonary veins and small indeterminate vessels [20]. In fact, the intimal thickening of these vessels was more severe than in arterioles, sharing a similar but milder pattern with primary pulmonary venoocclusive disease. Above all, the hemodynamic severity of PH, as characterized by PVR, correlated best with the intimal thickness of the venous and the indeterminate vessels [20]. These findings challenge the classical hemodynamic compartmentation of the vascular tree and, more importantly, plead for further research on the mechanisms that trigger and sustain intimal thickening in all components of small vasculature of the lung.

## 3. The Cellular Bases of PH-LHD

A retrograde increase in capillary pressure affects the function of pulmonary endothelial cells (ECs) [21]. Capillary injury and subsequent inflammation (Figure 1) induce activation and proliferation responses of fibroblasts/myofibroblast by means of neurohumoral mediators. This results in collagen deposition and thickening of the alveolar septa [22,23]. In parallel to the infiltration of inflammatory cells in the perivascular regions, there is also a thickening of the intima and media of arteries and arterioles, indeterminate vessels, and venules [24,25]. Therefore, ECs seem to be the primary cell type affected by PH, responsible for dysfunctional changes of the endothelium. Beyond the proliferative response, there is also an increased secretion of cytokines and growth factors (e.g., transforming growth factor alpha 1, TGF-α1), vascular endothelial growth factor (VEGF), interleukin 1 (IL-1)), an increased production of vasoconstrictors such as endothelin-1, and a decreased generation of vasodilators such as nitric oxide (NO) [26,27,28]. Damaged ECs also induce recruitment, proliferation, and subsequent differentiation of smooth muscle cells (SMCs). Moreover, the synthetic/proliferative SMCs, which have migrated to the intima, are capable of generating proinflammatory molecules that promote leukocyte infiltration of the arterial wall [28]. Fibroblasts in the pulmonary vascular wall play specific roles in the response to injury, including migration, proliferation, synthesis of intravascular connective tissue, contraction, cytokine production, and, most importantly, trans-differentiation into other types of cells (e.g., SMC) [29]. Inflammatory cells may also play a role in PH-LHD, as suggested by a rodent model of heart failure induced by supravalvular aortic stenosis in which the development of pulmonary fibrosis correlated with the degree of leukocyte infiltration of the lung [22].

## 4. The Molecular Pathways of PH-LHD

Most of the current knowledge in this field comes from Group 1-PAH and includes endothelial dysfunction, altered mitochondrial function, oxidative metabolism, inflammation, and immune alterations (Figure 1). Some of these mechanisms are presumed to be common among all PH groups, including PH-LHD. Major mechanisms are summarized below, with different supportive evidence for PH-LHD.

### 4.1. Endothelial Dysfunction and Endothelial Vasoactive Mediators

Endothelial dysfunction is caused by the altered production of various endothelial vasoactive mediators. Impairment of the endothelium-derived nitric oxide (eNOS), prostacyclin, angiopoietin-1, and endothelin-1 (ET-1) pathways are involved in all forms of PH [30].

Endothelium-derived nitric oxide (eNO) is essential for vascular homeostasis [31]. Although the role of eNO in the pathogenesis of PH remains poorly delineated, it is known that the NO availability is at least partially responsible for reduced pulmonary vasorelaxation. NO is primarily synthesized in the endothelium from L-arginine and oxygen by the endothelial NO synthase (eNOS, NOS3) triggered by mechanical stimuli or endogenous mediators, such as acetylcholine and bradykinin. NO diffuses to subjacent SMCs. The effects of this gaseous vasodilator are largely exerted by the activation of its molecular receptor, soluble guanylate cyclase (sGC), and the subsequent generation of cGMP through the action of phosphodiesterase type-5 (PDE-5). Hence, reduced l-arginine bioavailability or increased methylarginine production may reduce the production of NO and directly lead to endothelial dysfunction affecting pulmonary vascular reactivity [32]. Both a decrease of NO production and a reduction of vasodilator responsiveness to NO occur in PH-LHD [33]. However, all multicentric clinical trials in LHD targeted to increase eNO have shown negative or neutral results, whether using inhaled inorganic nitrite [34], PDE-5 inhibitors [35,36], or sGC stimulators [37,38,39] (Table 1, Figure 1).

The prostacyclin (PGI2) is a potent pulmonary vasodilator and platelet antiaggregant. PGI2 is synthesized from cyclooxygenase via the arachidonic pathway in the vascular endothelium and protects against pulmonary vasoconstriction and remodeling in response to various stimuli. For instance, PGI2 overexpression protects mice from chronic hypoxia–induced PH [47]. On the contrary, PGI2 receptor–deficient mice develop severe PH when exposed to chronic hypoxia. A decrease in PGI2, described in patients with different types of PH, may explain pulmonary vasoconstriction, smooth muscle cell proliferation, and enhanced coagulation [30]. However, when tested in a large-scale multicentric clinical trial, synthetic prostanoids significantly worsened survival compared to placebo in a large-scale clinical trial of patients with heart failure with reduced ejection fraction (HFrEF) [40] (Table 1).

Angiopoietin-1 (Ang-1) is an essential factor for lung vascular development [48]. Produced by SMCs and precursor pericytes, Ang-1 stabilizes the development of blood vessels by recruiting muscle cells through migration and division, inducing the creation of mature arterial structures from endothelial tubes. Its receptor, Tie2, is present only on the vascular endothelium. During organ development, the ligand–receptor interactions between smooth-muscle cells and endothelium-specific Tie2, induce the proliferation of muscle cells around the perivascular network [49]. After development, angiopoietin-1 is expressed at a minimally detectable level in the human lung, whereas the Tie2 receptor, believed to regulate vascular maintenance, remains constitutively expressed. An overexpression of Ang-1 with the activation of its receptor has been described in non-familial PH [50]. Moreover, levels of Ang-1 in the lung correlate directly with the severity of pulmonary hypertension in patients, irrespective of the disease [51]. Studies conducted in animal models also support the impact of Ang-1 overexpression on alterations of pulmonary SMCs described in PH patients [50].

Endothelin-1 (ET-1) is a potent vasoconstrictor widely distributed in the human endothelium with platelet-aggregating properties [52]. In patients with PH, including PH-LHD patients, ET-1 is overexpressed [53]. ET-1 binds to endothelin receptor type A (ETA) and endothelin receptor type B (ETB) expressed mainly on vascular smooth muscle cells (VSMCs) and cardiac myocytes, activating phospholipase C, which generates diacylglycerol and inositol triphosphate. These secondary messengers trigger the release of intracellular calcium, which activates myosin light chain kinase, phosphorylation of myosin light chain, and induce vasoconstriction [30]. ET-1 also activates the RhoA/Rho kinase pathway, which leads to calcium sensitization and sustained vasoconstriction [54]. ETB receptors are expressed on vascular endothelial cells, and upon binding of ET-1, promote production of NO and prostacyclin, resulting in vasodilation [55]. Unfortunately, potent ET antagonists such as bosentan, darusentan, and macitentan have all shown either neutral or harmful results when specifically tested in the field of PH-LHD [41,42,45,46] (Table 1).

The ras-homolog family member A/Rho associated coiled-coil containing protein kinase (RhoA/ROCK) pathway seems to be specifically implicated in PH-LHD [24,56,57]. ROCKs produce sustained vasoconstriction due to calcium sensitization resulting from their interactions with myosin light chain and myosin phosphatase target subunit-1 [58]. In PH, ROCK activation increases VSMC proliferation, inflammatory cell migration, platelet activation, ROS production, and endothelial dysfunction [58]. In a rat model of PH-LHD, ROCK inhibition with fasudil was shown to increase the pulmonary expression of eNOS, while mitigating PH-LHD, decreasing the medial thickness of the pulmonary artery by 50%, lowering the mPAP by 56%, and reducing the right ventricular hypertrophy by 30% [59]. Preliminary data from an open label small clinical study suggest that fasudil could also have beneficial hemodynamic effects in patients with heart failure, preserved ejection fraction, and cPC-PH [60].

### 4.2. Alterations in Platelet Bioenergetics and Mitochondria Dysregulation

Mediated by mitochondrial dysfunction, abnormal cell metabolism in multiple tissues contributes to PH both in animal models and in humans PAH [61]. In pulmonary vascular cells, a metabolic switch to aerobic glycolysis confers resistance to apoptosis, cellular hyperproliferation, and vascular remodeling. Under aerobic glycolysis, pulmonary vascular cells also exhibit increased mitochondrial reserve respiratory capacity, which depends on fatty acid oxidation increment and correlates with PH hemodynamic changes [62]. Moreover, this dysregulation also includes alterations in the pentose phosphate pathway, increases reliance on metabolic activities of HIF-2α and ROS signaling, and profound alterations in iron metabolism [63]. Thus, dysregulation of mitochondrial metabolism represents a key feature in the pathobiology of PH, at least in PAH. In fact, a comprehensive hypothesis of PAH focuses on the mitochondria as the epicenter of metabolic control and dysregulation, emphasizing the metabolic shift of glycolysis which involves ATP generation, glucose, fatty acids, and ketone bodies [64]. Whether mitochondrial dysregulation takes place in other PH forms is unknown, as well as the role of the mitochondria in PH-LHD. However, alterations in platelet bioenergetics, a known proxy of mitochondrial function changes, have been recently reported in PH-LHD [65]. These alterations are characterized by an increase in respiratory capacity, similar to PAH subjects, but not by an increase in glycolysis. Further, nitrite has shown no effect on the mitochondrial respiration of platelets [65].

Finally, intracellular lipid chaperones, fatty acid binding proteins (FABPs), coordinate lipid responses in cells and are strongly linked to metabolic and inflammatory pathways [66]. A particular type of FABP, a cardiac-specific cytoplasmic protein, has been reported as a marker for ongoing myocardial damage, with a potential prognostic value of adverse cardiac events. This protein, absent in Group-1 PAH, has been recently proposed as major indicator of PH-LHD [67]. This observation is probably due to the specificity of this molecule for left-heart damage, but also supports distinctive metabolic dysfunctions in PH-LHD.

### 4.3. Reactive Oxygen Species

Reactive oxygen species (ROS) play a major role in PH [68,69]. In fact, pulmonary vascular dysfunction can be promoted by increasing ROS production or decreasing ROS catabolism. In endothelial cells, ROS cause endothelial dysfunction by promoting proliferation, decreasing NO production, and inducing the release of vasoactive mediators. In SMCs, oxidative stress caused by ROS induces contraction and a switch to a synthetic phenotype, characterized by increased VSMC migration and proliferation, hypertrophy, and extracellular matrix protein deposition. These SMC alterations are mediated by an increase of the intracellular free Ca2+ and a reduction of the expression of contractile phenotypic markers, while enhancing simultaneously the expression of proliferative markers and growth factors. ROS can also trigger the proliferation of fibroblasts. These changes result in vascular remodeling and the development of PH [68]. Additionally, some specific redox changes resulting from oxidation, S-nitrosylation, and S-glutathionylation modulate membrane receptor and ion channel activity in PH [70,71]. Ion channels and specifically voltage-gated potassium channels seem to play an important role in the regulation of pulmonary vascular tone, and their inhibition has been described to be involved in PAH [72,73]. To date, these ubiquitous effects of ROS on all three layers of the pulmonary vessels have not been specifically addressed in PH-LHD.

### 4.4. Inflammation and Immunity

Amongst all groups of PH, it seems that PH-LHD has the highest impact on the inflammatory processes and matrix remodeling of the pulmonary vascular tree [67]. The accumulation of mast cells in the lungs of patients with PH-VHD caused by mitral stenosis was described decades ago and was associated with the muscularization of pulmonary vessels [74]. These observations suggested a functional role for mast cells in lung vascular remodeling. Later studies have confirmed immune cell abundance and infiltration in lung vascular lesions and in remodeled vessels [75], not only in Group 1-PAH [76], but also in PH-LHD [77]. These inflammatory changes are followed by stimulation of the adaptive immune system driven by interleukin 6 and signal transducer and activator of transcription 3 (IL6-STAT3) pathway [78]. In fact, in a preclinical rat model of PH-LHD, generated by combining supracoronary aortic banding and chronic inflammation-triggered by metabolic syndrome induction, the relevance of IL6-STAT3 pathway has been recently validated [79]. In a rat model of PH-LHD, mast cells have been shown to induce the release of serotonin and histidine, key role mediators in the vasoconstriction of pulmonary arteries and veins as well as in the proliferation of SMCs by activating the renin-angiotensin system [80]. Mast cells also produce collagen-cleaving matrix metallopeptidase 13 (MMP-13), a platelet-derived growth factor and transforming growth factor that stimulates SMC proliferation [81]. Moreover, mast cells also secrete activin A that promotes the proliferation of SMCs [82].

Recently, the growth differentiation factor 15 (GDF-15) and the soluble urokinase-type plasminogen activator receptor (suPAR), markers of inflammation and involved in the regulation of cell repairmen and growth (Figure 1), have been related to PH-LHD [67]. Unfortunately, the abundance of candidate pathways would require a detailed dissection of signaling events and large human datasets are necessary to identify and validate these and other potential immunological mechanisms involved in PH-LDH [83].

## 5. The Genetics of PH-LHD

Genetic predisposition, environment, and epigenetics are key mechanisms involved in the genesis of PH (Figure 1). Moreover, the factors responsible for the progression of PH remain poorly defined. It is believed that in most types of PH, overt disease involves multiple events on a background of genetic predisposition [25].

Most of current knowledge in this area comes again from the field of PAH [84,85,86,87,88,89]. Variants in bone morphogenetic protein type II receptor (BMPR2) are well-known determinants of PAH severity. The *BMPR2* gene encodes the BMPR-II survival regulator of ECs in the pulmonary artery. Mutations on *BMPR2* gene (Table 2) lead to a loss of BMPR2 signaling which predisposes to apoptosis of the endothelial cells. This is believed to be the primordial mechanism that initiates PAH [86,90,91,92,93,94]. Although *BMPR2* mutations are the most common inherited risk factors for PAH, only the 20% of carriers develop the disease [95]. Therefore, other genetic (Table 2) and environmental factors such as inflammation must be involved in vascular remodeling [90]. Amongst other genetic factors, mutations in more than 30 genes have been related to Group 1-PAH [84,86,96,97,98,99]. In addition to these causal rare sequence variants, disease penetrance and progression has been associated with variants in genetic modifiers [99,100,101,102,103,104]. A systematic review of genetic mutations in PAH can be found in [86].

While the genetic background of PAH has been tested in other PH forms such as chronic thromboembolic pulmonary hypertension (CTEPH) [105] and hereditary hemorrhagic telangiectasia [106], the genetic variants of PH-LHD have been barely studied [107,108]. It is assumed that *BMPR2*, *ALK1*, and other gene variants associated with PAH are only responsible for a minute fraction of patients with PH-LHD [109]. Recently, a genome-wide association study has identified gene variants in NADPH oxidase 3 (*NOX3*) and *TBX4* genes for unclassified non-idiopathic PH in an Eastern Chinese population [110]. Very few studies have been focused on candidate gene/variant analysis in PH-LHD. Due to its absence in the iPC-PH phenotype, a missense variant (rs1799983) in the endothelial NOS (*NOS3*) gene has been related to pulmonary vascular remodeling in cPC-PH [111]. However, this *NOS3* gene variant is a well-known polymorphism that produces an amino acid change from glutamic acid to asparagine with a global minor allele frequency (MAF) of 0.18 and it is currently classified as benign (ClinVar database). A repeat length polymorphism in the promoter region of the serotonin transporter solute carrier family 6 member 4 (*SLC6A4*, also known as *5-HTT* and *SERT*)) gene has been related to mPAP in heart failure [112]. The *SLC6A4* polymorphism (*SLC6A4* c.-1941_-1899indel, current annotation) consists of a 43-bp insertion or deletion involving repeat elements that affects protein activity. This variant is classified as pathogenic in ClinVar linked to behavior disorders but the correlation with pulmonary hypertension is unconfirmed [112].

The most encouraging results on the genetics of PH-LHD come from Assad studies [9,113]. The authors analyzed pre-existing genotyping data from the Illumina Infinium Human Exome BeadChip in populations with PAH, cPC-PH, and iPC-PH. In addition, they also exploited the Genotype-Tissue Expression (GTEx) database, targeting quantitative trait loci (eQTL) and their underlying genes [9]. Their study reported 141 SNPs that were differentially expressed in PAH and cPC-PH but not in iPC-PH. Amongst them, a missense variant in the collagen type XVIII alpha 1 chain (*COL18A1*) gene (rs62000962, V661I, MAF = 0.11, classified as benign in ClinVar) and another missense variant in the mitochondrial elongation factor 2 gene (*MIEF2*)—also known as *SMCR7* (rs12603700, G324E, MAF 0.12, not reported in ClinVar)—have been first putatively associated with PAH [114,115] and recently identified in patients with cPC-PH.

Also, the overrepresentation of lung-relevant functional pathways such as actin binding, extracellular matrix, basement membrane, transferase activity, pre-ribosome structure, and the major histocompatibility complex were also reported. Overall, the study supports the existence of genetic abnormalities in pathways that are highly active in the lungs in patients with PH-LHD, particularly prevalent in cPC-PH. Moreover, the study supports the possibility of common pathophysiological mechanisms between PAH and cPC-PH, [9] although none of the genes found genetically altered in PAH [86] are described in this study. In addition, based on their potential role on inflammatory processes, matrix remodeling and mitochondria dysregulation, some of the main mechanisms specifically involved in PH-LHD, those overrepresented genes can reclassified (David database: david.ncifcrf.gov/) as in Table 3, and they could be taken into consideration for further PH-LHD investigations.

## 6. The Epigenetics of PH-LHD

In addition to unknown genetic variants, the variability of PH-LHD may be related to epigenetic changes that regulate the vascular responses. In fact, the role of epigenetic regulation (methylation, histone, and chromatin modifications, and non-coding RNA, e.g., miRNAs, lncRNAs) in PH is being clarified [116]. However, the exact mechanisms involved remain largely unknown. Methylation of the superoxide dismutase-2 (*SOD2*) gene has been related to PAH [117]. The *SOD2* gene encodes for a member of the SOD family that controls the production of endogenous H_2_O_2_, which is a key factor in mitochondrial metabolism [118]. A decrease in *SOD2* expression has been shown in PAH patients through a hypermethylation at the enhancer region of intron 2 and promoter region. This epigenetic silencing of SOD2 contributes to the activation of hypoxia-inducible factor 1 and creates a pro-proliferative, apoptosis-resistant state [117]. Moreover, a global DNA methylation and histone acetylation reduction has been described in PH triggered for long-term hypoxia induced by altitude. These epigenetic alterations may lead to SMC proliferation and vessel remodeling in the pulmonary arteries [119]. More recently, a genome-wide DNA methylation study in PAH patients reported a relationship between ATP binding cassette subfamily A member 1 gene (*ABCA1*) downregulation and lipid metabolism [120]. Additionally, a decrease in granulysin gene (*GNLY*) demethylation has been described in the blood and lungs in PVOD patients [121]. The role of these mechanisms in PH-LHD remains unclear.

Aberrant histone acetylation-deacetylations have been described in animal PH models and PAH cells [116]. Increased H3 and H4 acetylation of *eNOS* expression increases the levels of eNOS in pulmonary arterial endothelial cells [122]. Increased expression and activity of class I deacetylases promote the suppression of pro-inflammatory mediators in fibroblasts [123]. Also, a reduction of the mitochondria-localized deacetylase sirtuin 3 gene (*SIRT3*) expression suppresses mitochondrial function, inhibits apoptosis, and activates several pulmonary hypertension-related transcription factors [124]. Besides, increased expression of a histone methyltransferase, the enhancer of zeste homologue 2, is found in proliferating PASMCs and associated with the progression of PH [125].

More than 10 years ago, non-coding RNA and specifically microRNAs (miRNAs) have been known to be involved in the development and progression of PH [126]. MiRNA are believed to coordinate and regulate multiple disease pathways in the pulmonary vasculature. More than 100 studies have reported associations between more than 30 miRNAs and different non-Group 2 forms of PH [127,128,129,130,131]. Amongst them, the most promising results have been identified in PAH: miR-21 [132], miR-17/92 [133], miR-204 [134], miR-130/301 [131], and Let-7d-3p [135]. Very few miRNAs have been specifically investigated in PH-LHD. The miR-204, which is downregulated in SMCs of PAH patients, has not been detected in PH-LHD patients, supporting differences in muscle-specific pathobiology between PAH and PH-LHD [136]. On the contrary, plasma levels of miR-206 do correlate with the clinical worsening of PH-LHD patients [137]. MiR-206 belongs to the miR-1 family, primarily expressed in skeletal muscle. Nevertheless, miR-206 expression has been detected in SMCs of pulmonary arteries and cardiomyocytes. A reduction in miR-206 seems to increase proliferation and reduce apoptosis of SMCs. Although the regulation of miR-206 in PH-LHD remains largely unknown, the reported data supports that miR-206 regulates the cellular features of both cardiomyocytes and SMCs, indirectly suggesting a potentially relevant role in PH-LHD [137]. In addition, based on their role on regulation, enhancement, and/or repressor of different biological pathways, several miRNAs may hypothetically play a role in PH-LHD (Table 4).

## 7. Conclusions

The prognosis of patients with PH-LHD is close to patients with PAH or even worse. As summarized above, clinical trials testing drugs that are proven to be highly effective in PAH, have shown either harmful or negative results when the drugs are tested in PH-LHD. Therefore, the identification of new therapeutic targets is essential. Based on the current evidence herein described, some novel strategies deserve further investigation. Overexpressing miR-206 would potentially suppress *EDN1* gene expression, decreasing endothelin-1, which in turn could inhibit ROCK and increase of eNOS pulmonary expression. This could mitigate the VSMC proliferation, inflammatory cell migration, platelet activation, ROS production, and endothelial dysfunction. Testing the potential therapeutic value of this and additional molecular pathways deserves active pre-clinical research in PH-LHD.

Furthermore, due to the well-known limitations of the current functional and hemodynamic criteria, patients with PH-LHD would benefit from a personalized biological stratification of their disease. Both these goals require a comprehensive translational approach that must begin by deciphering the similarities and singularities of PH-LHD and other forms of the disease. Thus far, only limited information regarding the genetic and molecular features of PH-LHD has been exploited. Molecular investigation based on -omics (genomics, transcriptomics, epigenomics, metabolomics, and proteomics) in deeply-phenotyped cohorts may provide important mechanistic information, novel biomarkers and, most importantly, potential therapeutic targets of the disease.

## Figures and Tables

**Figure 1 ijms-20-05884-f001:**
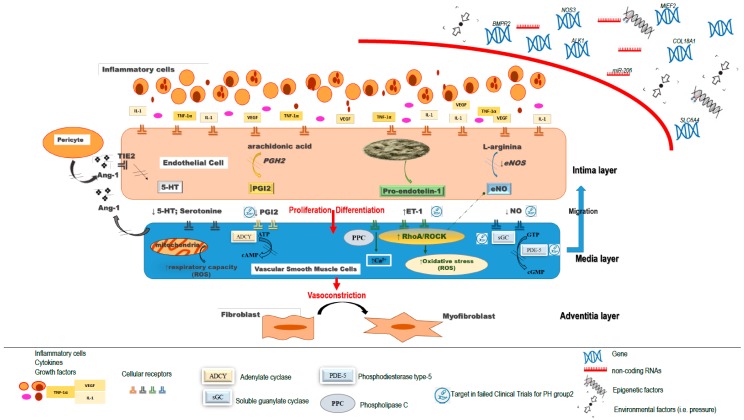
Representation at the vessel wall level of major mechanisms involved in pulmonary hypertension forms with different supportive evidence for left heart disease.

**Table 1 ijms-20-05884-t001:** Summary of multicentric clinical trials in left heart disease (LHD) patients suffering from pulmonary hypertension (PH).

Pathway	Drug	N	Condition	PH-Focused	Main Finding	ACRONYM/Ref
**eNO**	Inorganic Nitrate	105	HFpEF	no	neutral	INDIE-HFpEF [34]
**PGI2**	Epoprostenol	471	HFrEF	no	harmful	FIRST [40]
**PDE-5**	Sildenafil	216	HFrEF	no	neutral	RELAX [36]
	Sildenafil	200	VHD	yes	harmful	SIOVAC [35]
**sGC**	Riociguat	201	HFrEF	yes	neutral	LEPHT [37]
	Vericiguat	456	HFrEF	no	neutral	SOCRATES-REDUCED [38]
	Vericiguat	477	HFpEF	no	neutral	SOCRATES-PRESERVED [39]
**ET-1**	Darusentan	157	HFrEF	yes	neutral/harmful	HEAT [41]
	Darusentan	642	HFrEF	yes	neutral/harmful	EARTH [42]
	Bosentan	369	HFrEF	yes	harmful	REACH-1 [43]
	Bosentan	87	HFrEF	yes	neutral/harmful	NA [44]
	Bosentan	1613	HFrEF	yes	neutral/harmful	ENABLE [45]
	Macicentan	63	HFrEF & pEF	no	harmful	MELODY-1 [46]

eNO: endothelium-derived nitric oxide; PGI2: prostacyclin; PDE-5: phosphodiesterase type-5; sGC: soluble guanylate cyclase; ET-1: Endothelin-1; HFpEF: heart failure with preserved ejection fraction; HFrEF: heart failure with reduced ejection fraction; PH-focused: whether patients were specifically screened for PH or not.

**Table 2 ijms-20-05884-t002:** Summary of variants described in major genes associated with pulmonary arterial hypertension (PAH) forms (ClinVar, https://www.ncbi.nlm.nih.gov/clinvar/) (August 2019).

	ClinVar Classification				
Gene	P: Pathogenic	LP: Likely Pathogenic	Uncertain Significance	Benign/Likely Benign	PH Form (Number of Causal Variants; P/LP)
*BMPR2*	405	8	83	70	PAH (387); PVOD (2); PAH-CHD (2)
*ENG*	187	32	137	62	PAH (5)
*SMAD9*	37	3	67	63	PAH (3)
*CAV1*	27	1	6	6	PAH (3)
*KCNK3*	17	1	21	6	PAH (7)
*ALK1/ACVRL1*	130	30	96	52	PAH-HHT (27); PAH (5)
*TBX4*	23	5	24	32	PAH-CHD (10); PAH (1)

PAH: pulmonary arterial hypertension; PVOD: Pulmonary venoocclusive disease; PAH-CHD: Pulmonary arterial hypertension associated with congenital heart disease; PAH-HHT: Pulmonary arterial hypertension related to hereditary hemorrhagic telangiectasia.

**Table 3 ijms-20-05884-t003:** Reclassification of genes found overrepresented in PAH and cPC-PH vs. iPC-PH [9].

Biological Term (GeneOntology, UniProtKB)	Gene Official Symbol/Function Associated (www.ncbi.nlm.nih.gov/gene)
Extracellular matrix	*FREM1*/Craniofacial and renal development regulator
	*COL4A3*/Major structural component of basement membranes
	*COL18A1*/Inhibitor of angiogenesis and tumor growth
	*LAMA5*/Noncollagenous basement membrane component
	*ADAMTS7*/Regulator of vascular smooth muscle cell migration
Immune system/Inflammation	*FREM1* short isoform/Co-receptor of the interleukin 1
	*UMODL1*/Autoimmune diseases
	*OR2C3*/Immune system regulator
	*RALY*/Ribonucleoprotein involved in autoimmune responses
	*FPR1*/Host defense and inflammation component
	*DPA1*/Class II major histocompatibility complex
Mitochondrial/Oxidative stress	*CASP2*/Regulator of stress-induced signaling pathways
	*NEK5*/Mitochondrial mediated cell death and respiration
	*SMCR7*/Mitochondrial fission machinery component
	*ZNF3*/Oxidative stress response

**Table 4 ijms-20-05884-t004:** Top five scored miRNAs predicted to target major genes potentially involved in PH-LHD (miRDB database [138]).

Targeted Gene Symbol	Role in PH-LHD	miRNA Name
*NOS3*	NOS3 polymorphism in PH-LHD [111]	miR-154-5p, miR-1303, miR-1206, miR-377-3p, miR-668-5p
*COL18A1*	SNPs differentially expressed in cPC-PH [9]	miR-1972, miR-6762-3p, miR-5580-5p, miR-3118, miR-134-5p
*MIEF2*	SNPs differentially expressed in cPC-PH [9]	miR-3128, miR-6785-5p, miR-4667-3p, miR-922, miR-629-5p
*SLC6A4*	repeat length polymorphism associated to PH-HF [112]	miR-4775, miR-1250-3p, let-7b-3p ^1^, miR-98-3p, let-7f-1-3p ^1^
*RHOA*	RhoA/ROCK pathway specifically involved in PH-LHD [24]	miR-451b, miR-582-3p, miR-3646, miR-1207-3p, miR-6815-3p
*ROCK2*	RhoA/ROCK pathway specifically involved in PH-LHD [24]	miR-5011-5p, miR-3163, miR-190a-3p, miR-30d-3p, miR-30e-3p
*EDN1*	endothelin-1 activates the RhoA/Rho kinase pathway [54]	miR-7113-3p, miR-4287, miR-651-3p, miR-671-5p, miR-206 ^2^
*EDNRA*	encodes for endothelin receptor type A; endothelin-1 activates the RhoA/Rho kinase pathway [54]	miR-148b-3p, miR-335-3p, miR-3671, miR-3686, miR-607
*EDNRB*	encodes for endothelin receptor type B; endothelin-1 activates the RhoA/Rho kinase pathway [54]	miR-30e-5p, miR-19b-2-5p, miR-4282, miR-19b-1-5p, miR-2052
*IL6*	IL6 drives adaptive immune system (mast cells) in PH-LHD [78]	miR-302d-3p, miR-11181-5p, miR-4256, miR-196a-1-3p, miR-548c-3p ^3^
*GDF15*	inflammation markers, correlated with PH-LHD [67]	miR-499b-5p, miR-6721-5p, miR-1324, miR-6740-5p, miR-374a-3p
*PLAU*	encodes for suPAR, correlated with PH-LHD [67]	miR-6131, miR-5692, miR-193a-3p, miR-193b-3p ^4^, miR-8485

^1^ Left-7 miRNA family members are particularly active in CTEPH [139]. ^2^ Circulating miR-206 levels correlate with PH-LHD [137]. ^3^ Dysregulated in porcine PH shunt model [140]. ^4^ Downregulated in the lung and serum of PAH patients and PH rodents [141].

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
