# Peer review of "The Biological Bases of Group 2 Pulmonary Hypertension"

_ijms, 2019, doi:10.3390/ijms20235884_

Round 1
Reviewer 1 Report
The present manuscript provides up-dated comprehensive review regarding cellular and molecular basis for pulmonary hypertension (PH), with a special focus on Group 2 PH.
Major points
Illustration (Figure 1): (1) There are some discrepancy between the title and the content of the figures. A large part of the figure shows molecular events of vascular smooth muscle cells. The title should be rephrase as as to be consistent with the contents of the figure. (2) More detailed description and explanation of the contents of figure in the legend would be helpful. Otherwise, it is unclear, for example, what "Clinical trials main finding" in blue indicates, etc. (3) The figure appears to illustrate general mechanism of the regulation of contraction/relaxation of vascular smooth muscle, with a endothelium-dependent mechanism. It is unclear what events are relevant to pathogenesis of Group 2-PH. An illustration that summarized the whole contents of the present manuscript, regarding the cellular, molecular, energetic, genetic and epigenetic events/mechanisms underlying pathogenesis and pathophysiology relevant to Group 2-PH is recommended to be shown to make the Group 2-PH-relevant points emphasized and clearer. The events/mechanisms that has no information so far, such as genetic predisposition, could be illustrated/shown as "no information", as it is considered to be important information that no information is available for certain event/mechanism. However, this comment is totally discretionary to authors.
Minor points
"A highly prevalent", a phrase in the first sentence of the Abstract, is somewhat counter-intuitive for describing PH. Did authors intend to emphasize the fact that the prevalence of PH has been reported to be around 1% of the world population as in the reference 1? Some amendment is recommended to support such an apparent counter-intuitive expression. "type 2" in Keywords should be "Group 2". L.140-141: The statement "NO is at least partially responsible for reduced pulmonary vasorelaxation" is recommend to be rephrase. L.149-151: There seems to be some syntax error. Please take a reconsideration for the sentence. L.158 and other parts: "PGI2" is usually used to represent prostacyclin. L.178: "Animal findings" is somewhat inappropriate expression. L.191: It should be "macitentan". L.308: "Assad and cols" appears to be inappropriate for scientific writing.

Author Response
Response to Review1
The present manuscript provides up-dated comprehensive review regarding cellular and molecular basis for pulmonary hypertension (PH), with a special focus on Group 2 PH.
Major points
Point 1- Illustration (Figure 1): (1) There are some discrepancy between the title and the content of the figures. A large part of the figure shows molecular events of vascular smooth muscle cells. The title should be rephrase as as to be consistent with the contents of the figure. (2) More detailed description and explanation of the contents of figure in the legend would be helpful. Otherwise, it is unclear, for example, what "Clinical trials main finding" in blue indicates, etc. (3) The figure appears to illustrate general mechanism of the regulation of contraction/relaxation of vascular smooth muscle, with a endothelium-dependent mechanism. It is unclear what events are relevant to pathogenesis of Group 2-PH. An illustration that summarized the whole contents of the present manuscript, regarding the cellular, molecular, energetic, genetic and epigenetic events/mechanisms underlying pathogenesis and pathophysiology relevant to Group 2-PH is recommended to be shown to make the Group 2-PH-relevant points emphasized and clearer. The events/mechanisms that has no information so far, such as genetic predisposition, could be illustrated/shown as "no information", as it is considered to be important information that no information is available for certain event/mechanism. However, this comment is totally discretionary to authors.
Answer-A new Figure 1 has been elaborated in order to illustrate general cellular, molecular, genetics and epigenetics events. We consider it could be used to summarize the whole article.
The title has been changed: Figure 1. Representation at the vessel wall level of major mechanisms involved in pulmonary hypertension forms with different supportive evidences for left heart disease.
Minor points
Point2-"A highly prevalent", a phrase in the first sentence of the Abstract, is somewhat counter-intuitive for describing PH. Did authors intend to emphasize the fact that the prevalence of PH has been reported to be around 1% of the world population as in the reference 1? Some amendment is recommended to support such an apparent counter-intuitive expression.
Answer-The phrase has been changed: "Pulmonary hypertension (PH) is a potentially fatal condition with a prevalence around 1% in the world population and most commonly caused by left heart disease (PH-LHD)".
Point 3-"type 2" in Keywords should be "Group 2".
Answer-Changed
Point 4-L.140-141: The statement "NO is at least partially responsible for reduced pulmonary vasorelaxation" is recommend to be rephrase.
Answer-The phrase has been changed: “it is known that the NO availability is at least partially responsible for reduced pulmonary vasorelaxation”
Point 5-L.149-151: There seems to be some syntax error. Please take a reconsideration for the sentence.
Answer-The phrase has been changed: “Both, a decrease of NO production and a reduction of vasodilator responsiveness to NO, occur in PH-LHD”
Point 6-L.158 and other parts: "PGI2" is usually used to represent prostacyclin.
Answer-Changed along the manuscript
Point 7-L.178: "Animal findings" is somewhat inappropriate expression.
Answer-Changed: Studies conducted in animal models
Point 8-L.191: It should be "macitentan".
Answer-Changed to macitentan
Point 9-L.308: "Assad and cols" appears to be inappropriate for scientific writing.
Answer-Changed to “The most encouraging results on the genetics of PH-LHD come from Assad studies”.
Reviewer 2 Report
The authors summarized in this review article the biological mechanism of group 2 pulmonary hypertension. Despite significant improvement in the understanding of biological mechanisms in group 1 pulmonary hypertension, little is known about group 2 pulmonary hypertension. The clinical therapeutic strategy for group 2 pulmonary hypertension is also required to be established. The review article is well written and the detailed biological mechanism is described. There are several concerns to be clarified.
Several terms might be used without abbreviation. For example, TGF-a1 and VEGF.
In section 5 (the genetics of PH-LHD), is it possible to minimize the description of group 1 pulmonary hypertension and more focus on group 2 genetics?
Is it possible to mention the clinical therapeutic strategy for group 2 pulmonary hypertension based on the current evidence described in this review article for the interest of clinicians?
Author Response
Responses to Review2
The authors summarized in this review article the biological mechanism of group 2 pulmonary hypertension. Despite significant improvement in the understanding of biological mechanisms in group 1 pulmonary hypertension, little is known about group 2 pulmonary hypertension. The clinical therapeutic strategy for group 2 pulmonary hypertension is also required to be established. The review article is well written and the detailed biological mechanism is described. There are several concerns to be clarified.
Point 1-Several terms might be used without abbreviation. For example, TGF-a1 and VEGF.
Answer-Changed abbreviations Transforming growth factor alpha 1(TGF-α1), Vascular endothelial growth factor (VEGF), Interleukin 1 (IL-1), endothelin receptor type A (ETA) and endothelin receptor type B (ETB), fatty acid binding proteins (FABPs)
Point 2-In section 5 (the genetics of PH-LHD), is it possible to minimize the description of group 1 pulmonary hypertension and more focus on group 2 genetics?
Answer-We have changed genetics section in agreement with the suggestion, reducing group-1 description and explaining with more details the group-2 genetics. Even more, a new table (Table 3) has been incorporated with a reclassification of genes found overrepresented in PAH and cPC-PH vs. iPC-PH from the Assad study in order to support potential genetic background for PH-LHD.
Point 3- Is it possible to mention the clinical therapeutic strategy for group 2 pulmonary hypertension based on the current evidence described in this review article for the interest of clinicians?
Answer-Due to the limited available information regarding the genetic and molecular features of PH-LHD, we have mentioned a potential therapeutic strategy to be further investigate, focused on miR-206. A paragraph regarding this point has been added at the conclussion section.